

# Germination of pecan seeds changes the microbial community

Jia Liu[1], Sumei Qiu[2], Liping Yang[2], Can Yang[2], Tingting Xue[1] and Yingdan Yuan[2]

[1] Department of Civil and Architecture and Engineering, Chuzhou University, Anhui, China
[2] College of Horticulture and Landscape Architecture, Yangzhou University, Yangzhou, China

## ABSTRACT

Endophytes are core of the plant-associated microbiome, and seed endophytes are closely related to the plant growth and development. Seed germination is an important part of pecan's life activities, but the composition and changes of microbes during different germination processes have not yet been revealed in pecan seeds. In order to deeply explore the characteristics of endophytes during the germination process of pecan, high-throughput sequencing was performed on seeds at four different germination stages. Findings of present study was found that the diversity and composition of microorganisms were different in different germination stages, and the microbial richness and diversity were highest in the seed endocarp break stage. It was speculated that the change of endophytes in pecan seeds was related to the germination stage. By evaluating the relationship between microbial communities, the core microbiota *Cyanobacteria*, *Proteobacteria* and *Actinobacteria* (bacterial) and *Anthophyta* and *Ascomycota* (fungal) core microbiota were identified in germinating pecan seeds. Finally, biomarkers in different germination processes of pecan seeds were identified by LEfSe analysis, among which *Proteobacteria*, *Gamma proteobacteria* and, *Cyanobacteria* and *Ascomycota* and *Sordariomycetes* were most abundant. Thus, this study will help to explore the interaction mechanism between pecan seeds and endophytes in different germination processes, and provide materials for the research and development of pecan seed endophytes.

## INTRODUCTION

Seeds are the transmission medium of microorganisms from mother plants to their offsprings, passing beneficial microorganisms from generation to generation. Seed-borne microbes are present in different tissues of the plant and provide a range of benefits to the host plant (*Johnston-Monje & Raizada, 2011*; *Lopez-Velasco et al., 2013*; *Shade, Jacques & Barret, 2017*). Microorganisms in seeds are diverse, including taxa such as bacteria, fungi, and archaea (*Simonin et al., 2022*), and there are about 1–1,000 species of bacteria and 1–150 species of fungi in a single seed, and the diversity varies greatly among samples across plant species (*Shearin et al., 2018*; *Truyens et al., 2015*). Numerous microorganisms exist in and on seeds, and these microorganisms play important roles in seed storage (*Qi et al., 2022a*, *2022b*), seed germination (*Goggin et al., 2015*), seedling growth (*Gao et al.,

Corresponding authors
Tingting Xue,
xuetingting1991@outlook.com
Yingdan Yuan, yyd@yzu.edu.cn

2020; *Walsh et al., 2021*) and plant health (*Hu et al., 2020*). Endophytes play a direct or indirect role in seed germination to promote the growth of the host plant (*Ahmad et al., 2020*). Studies have shown that seed endophytes vary by plant species (*Links et al., 2014*), genotype (*Adam et al., 2018*), seed maturation process (*Mano et al., 2006*) and environmental conditions (*Klaedtke et al., 2016*). Seed endophytes are derived from plant-associated endophytes that can influence plant microbial community structure and function (*Midha et al., 2016*).

The dominant phyla of bacteria in seeds are *Proteobacteria*, *Actinobacteria*, *Firmicutes* and *Bacteroidetes*, the dominant phyla of fungi are *Ascomycota* and *Basidiomycota*, and the dominant phyla of archaea are *Thaumarchaeota* and *Euryarchaeota* (*Simonin et al., 2022*; *Taffner et al., 2020*). Microorganisms widely present in seeds include *Pantoea*, *Pseudomonas*, *Rhizobium*, *Dothideomycetes*, *Tremellomycetes* and *Candidatus nitrososphaera* and other taxa. Some bacteria can parasitize plant seeds and they may be the creators of rhizosphere or internal bacterial communities early in plant development. These seed endophytes have been shown to have beneficial effects on the host plant, such as releasing seeds from dormancy, promotes seed germination, defends against pathogens and promotes seedling growth (*Khalaf & Raizada, 2018*; *Rahman et al., 2018*). *Cladosporium cladosporioides* acts as a seed endophyte with properties that promote seed germination and plant growth (*Qin, Pan & Yuan, 2016*). *Fan et al. (2016)* isolated *Arthrobacter* and *Bacillus megaterium* with growth-promoting ability, which can effectively improve the germination rate of tomato seeds, seedling height, fresh weight and dry weight of plants. Moreover, after inoculating rice with good functional strains screened, four strains were found to significantly increase the root length, root weight, stem length and plant fresh weight of rice seedlings (*Fan et al., 2016*). *Choi, Jeong & Kim (2022)* showed that *Capsella bursa-pastoris* seeds harbor bacterial endophytes that stimulate seedling growth, thereby potentially affecting seedling establishment.

*Carya illinoinensis* is an economic tree species of Juglandaceae, also known as American hickory and long hickory. It is native to North America and is one of the world's famous dry fruit tree species (*Guo et al., 2020*). As an important woody oil tree species, pecan nuts have high benefit and wide application, and have a very high value of promotion and application (*Guo et al., 2020*). Its fruit is large, the shell is thin, delicious and nutritious, and is deeply loved by consumers all over the world. It is well known that pecan nuts have large seeds, thin shells and many nutrients with less astringency and can be eaten freshly. As a fruit, pecan, which is also used as a seed, contains rich microbial resources that have not been fully tapped so far (*Wells, 2017*). At present, there are not many high-throughput sequencing on the study of endophytes in pecan nuts, and most of them are limited to the isolation of endophytes. For example, the isolation of endophytes in walnut fruits is used to ferment cotton stalks to hydrolyze sugar liquid to produce oil (*Zhang, Li & Xia, 2014*; *Zhang et al., 2014*).

We present the study on identifying the pecan seed microbiome during the germination process. The microbiome niche differentiation of different germination process in pecan is evaluated in the present study. We have focused here on two primary issues: (1) Which kinds of bacteria and fungi are responsible for the germination process in pecan seeds? (2)

What are the similarities and differences in the microbial community composition of pecan germination process? By understanding the above two questions, we can further select the excellent strains that can promote seed germination of plants and apply them to the promotion and cultivation of pecan.

## MATERIALS AND METHODS

### Sampling procedure

Fresh pecan seeds were used in the present experiment, which were provided by the Jiang Tao Family Farm, Lai 'an County, Anhui Province, China in November 2022, and the cultivar is "Pawnee". The seeds were kept at a 30 °C room temperature in Chuzhou University's Plant Physiology Laboratory. For ten days, pecan seeds were imbibed at room temperature. The seeds were then placed at a germination box covered with moist absorbent cotton and placed in a 30 °C constant temperature in full light incubator. The samples were taken four times: S1 (imbibed 10 days), S2 (after 5 days in the incubator), S3 (seed endocarp break) and S4 (seed radicle protrusion) during seed germination. At each stage of germination, we collected 5–10 seeds that germinated consistently and surface sterilized them with 2% NaClO (sodium hypochlorite) by shaking them for 4 min. In order to eliminate residues of NaClO that may interfere with the next steps, the seeds were rinsed five times in sterile MilliQ water. Finally, all samples were immediately frozen in liquid nitrogen. For microbial sequencing, the samples were stored at −80 °C in a refrigerator.

### DNA extraction and amplification

CTAB (cetyl trimethylammonium bromide) technique was used to extracted total genome DNA from seed samples (*Niemi et al., 2001*). We have evaluated DNA content and purity of 1% agarose gels, what's more, depending on the concentration, DNA was diluted to 1 ng·µL$^{-1}$ with sterile water. To generate the bacterial libraries, we used the 799F (5′-AACMGGATTAGATACCCKG-3′) and 1193R (5′-ACGTCATCCCCACCTTCC-3′) primers set with the unique 6-nt barcode at 5′ of the forward primer to amplify the V5–V7 region of the 16S rRNA gene for each sample. The construction of the fungal libraries was similar to the bacterial libraries, except that they were amplified using ITS1F (5′-CTTGGTCATTTAGAGGAAGTAA-3′) and ITS2 (5′-GCTGCGTTCTTCATCGATGC-3′) for the ITS2 region. All PCR reactions were performed using Phusion® High-Fidelity PCR Master Mix (New England Biolabs, Ipswich, MA, USA). One-load buffer (including SYB green) was mixed with PCR products at a 1:1 volume ratio, and electrophoretic detection was performed on 2% agarose gels. The PCR products were mixed in an equal density ratio. The mixed PCR products were then purified using the Qiagen Gel Extraction Kit (Qiagen, Hilden, Germany). In accordance with the manufacturer's recommendations, sequencing libraries were generated using the Truseq® DNA PCR-Free Sample Preparation Kit (Illumina, San Diego, CA, USA). Index codes were also added to the libraries. A Qubit@ 2.0 Fluorometer (Thermo Scientific, Waltham, MA, USA) and an Agilent Bioanalyzer 2100 system were used to evaluate the quality of the library.

An Illumina MiSeq PE300 platform was used to sequence the libraries and generate paired reads.

## DNA sequence analysis

All raw data from the 16S V5–V7 bacterial region and the fungal ITS1 region was processed by QIIME for quality-controlled process (V1.9.1, http://qiime.org/), and FLASH for paired reads (V1.2.7, http://ccb.jhu.edu/software/FLASH/) (*Bolyen et al., 2019*; *Magoč & Salzberg, 2011*). For bacteria and fungi, annotation was done by matching the Silva sequences with the UCHIME algorithm and Unite database (ITS: http://unite.ut.ee/) (UCHIME, https://www.drive5.com/usearch/manual/uchime_algo.html) (*Koljalg et al., 2013*; *Quast et al., 2012*).

## Statistical analysis

For the analysis of Alpha Diversity, four indices of species diversity are applied, including Observed Species, Chao1, Shannon, and Simpson. These indices were calculated using QIIME (Version 1.9.1) and visualized using R package ggplot2 (Version 2.15.3). One-way ANOVA was used to determine significant differences at a *P*-value of 0.05; if significant differences were detected, Duncan's *post hoc* test was used to identify the values that differed from the others. A beta diversity analysis was used to determine the differences in species complexity among samples, and beta diversity on Bray-Curtis was calculated using QIIME software (version 1.9.1).

## RESULTS

### Alpha and beta-diversity of pecan seed microbial communities

Pecan seeds were sequenced using the Illumina MiSeq high-throughput platform to determine the diversity of bacterial and fungal communities and their ASVs. The Good's coverage index of all samples was greater than 98.5%, indicating that the sequencing results could accurately reflect the true status of the bacterial and fungal communities within the samples. In terms of Shannon diversity index, bacteria had a higher diversity at S3 (seed endocarp break) stage of seed germination, whereas fungi had a higher diversity at S1 (imbibed 10 days) stage. Both the ACE index and the Chao1 index of seed endophyte microorganisms in the four stages had the same trend, and at S3 stage had the highest, indicating the most abundant species in the S3 stage. Additionally, the species richness at the S4 (seed radicle protrusion) stage in fungi was significantly lower than that of the other samples. It was found that the Pielou evenness index of bacteria was generally small, indicating that the species distribution of bacteria in pecan seeds was not uniform. However, the fungal index is close to one, indicating that fungal species are equally distributed, and at the S1 stage, species distribution is the most uniform (Table 1).

Principal components analysis (PCA) is a two-dimension reduction ordination analysis method based on Bray–Curtis distance for assessing the similarities and differences between bacterial and fungal communities (Fig. 1). In order to investigate the separation of different germination process, we performed PCA in our study. In bacterial PCA, 36.49%

**Table 1** α-diversity (mean ± SD) of pecan germination stages.

| Microbial | Sample ID | Shannon | Pielou evenness | Chao1 | ACE |
|-----------|-----------|---------|-----------------|-------|-----|
| Bacteria | S1 | 0.8 ± 0.15[a] | 0.16 ± 0.03[a] | 192.92 ± 20.15[b] | 193.59 ± 17.15[b] |
| | S2 | 0.77 ± 0.15[a] | 0.15 ± 0.03[a] | 197.94 ± 5.82[ab] | 190.23 ± 3.55[b] |
| | S3 | 1.38 ± 1.09[a] | 0.24 ± 0.18[a] | 284.55 ± 80.58[a] | 283.08 ± 80.87[a] |
| | S4 | 1.18 ± 0.75[a] | 0.22 ± 0.13[a] | 235.55 ± 35.95[ab] | 236.39 ± 37.26[ab] |
| Fungi | S1 | 3.54 ± 0.05[a] | 0.82 ± 0.03[a] | 83.34 ± 13.81[b] | 82.4 ± 11.26[b] |
| | S2 | 2.66 ± 0.38[b] | 0.59 ± 0.09[b] | 103.47 ± 25.93[ab] | 105.83 ± 23.57[ab] |
| | S3 | 2.67 ± 0.21[b] | 0.54 ± 0.05[b] | 149.59 ± 44.38[a] | 149.99 ± 42.97[a] |
| | S4 | 1.43 ± 0.42[c] | 0.38 ± 0.1[c] | 55.91 ± 13.31[b] | 58.66 ± 11.88[b] |

**Note:**
S1, Imbibed days; S2, incubator 5 days; S3, seed endocarp break; S4, seed radicle protrusion. Different lowercase letters indicate significant differences among pecan germination stages.

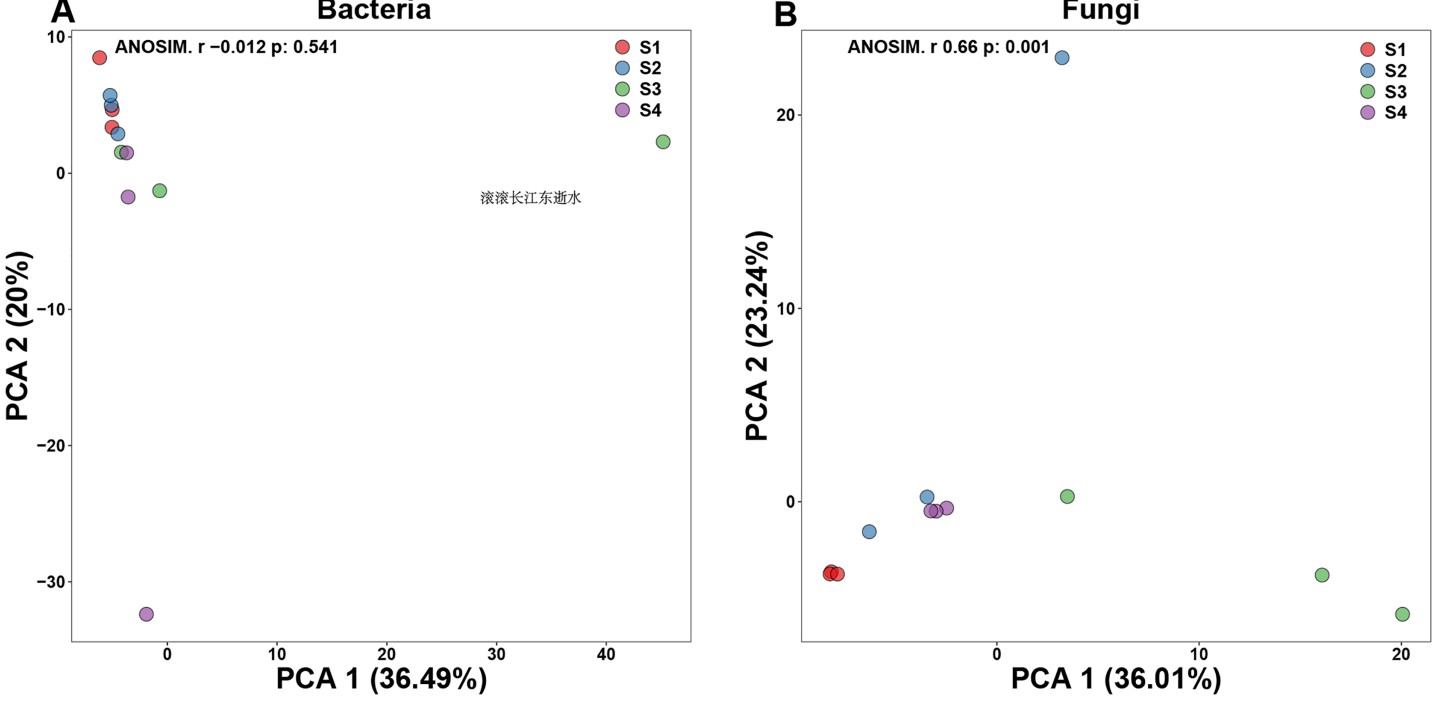

**Figure 1** **Beta-diversity in pecan seeds at different germination stage.** (A) Principal component analysis (PCA) of bacterial communities. (B) PCA of fungal communities.

of PC1 and 20% of the total variance were explained by PC2 (Fig. 1A). In fungal PCA, PC1 accounted for 36.01% and PC2 for 23.24% of the overall variance (Fig. 1B).

## Microbial community composition in pecan seed germination process

According to the analysis of ASV annotation results, different germination stages of pecan seeds bacteria include 12 phyla, 18 classes, 38 orders, 60 families, 82 genera and 62 species (Fig. 2). Among all sequences, the dominant bacterial phyla (relative abundance >1%) were *Cyanobacteria*, *Proteobacteria*, *Firmicutes*, *Actinobacteria*, *Bacteroidetes*, *Acidobacteria*, *Euryarchaeota*, *Deniococcus-Thermus*, *Spirochaetes*, *Ignavibacteriae*, *Parcubacteria*. It can
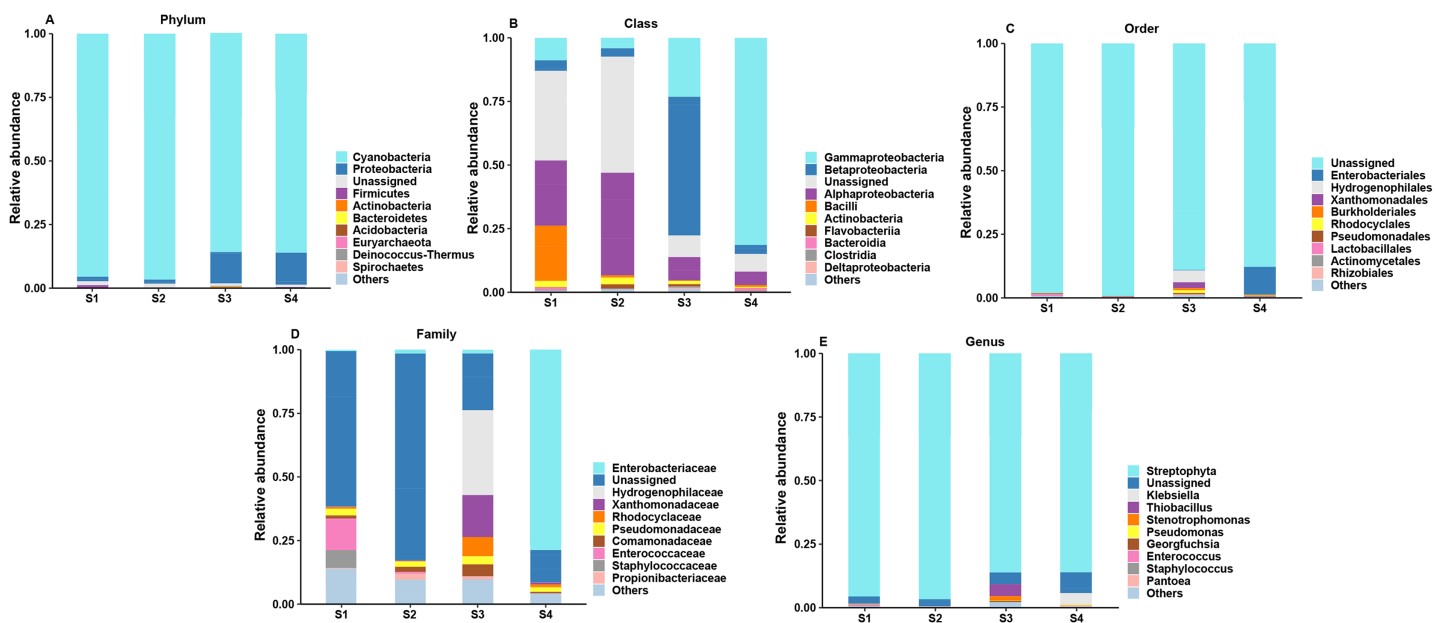

**Figure 2 Top 10 relative abundances of bacterial communities classified at different germination stages of pecan seeds as: (A) phylum, (B) class, (C) order, (D) family and (E) genus.**

be seen from Figs. 2A–2E that the endophytic bacteria in the seeds at the S3 and S4 stages are different from those in the previous two stages. At the phylum level, the two main phylums have undergone major changes, and *Cyanobacteria* in the first two stages. The relative abundance of the first stage is relatively higher, and in the latter two stages, the relative abundance of *Proteobacteria* increases, while the abundance of *Cyanobacteria* decreases. The S3 and S4 stages represent two important stages of seed germination, respectively, and they require different metabolites at the bacterial community level. So, the present study was found that in these two stages, their bacterial communities at different levels have great differences and changes.

However, the changes in the fungal relative abundance of the major families during germination are shown in Fig. 3. Pecan seeds fungi include seven phyla, 14 classes, 23 orders, 30 families, 35 genera and 22 species. Among all sequences, the dominant fungal phyla (relative abundance >1%) were *Anthophyta*, *Ascomycota*, *Basidiomycota*, *Cercozoa*, *Bacteroidetes*, *Chlorophyta*, *Chytridiomycota*, and *Mortierellomycota*. Among other species, we observed Juglandaceae, an endophyte closely related to pecan.

## Core and specific microbiome of pecan seeds

In order to assess the relationship between bacterial and fungal communities, the ASVs shared by different germination stages of pecan seeds were presented as an UpSet plot in Figs. 4A and 4B. UpSet plots are used for visualizing the number and overlap of different ASVs. As shown in Fig. 4A, all samples shared the majority of bacterial ASVs. There were 40 ASVs shared by S1, S3, and S4, 17 ASVs shared by S1 and S3 and 24 ASVs shared by S2 and S4. S3 stage was found to have the most unique ASVs compared with the other three stages. In all four germination stages, a total of 180 fungal ASVs was found. 64 ASVs

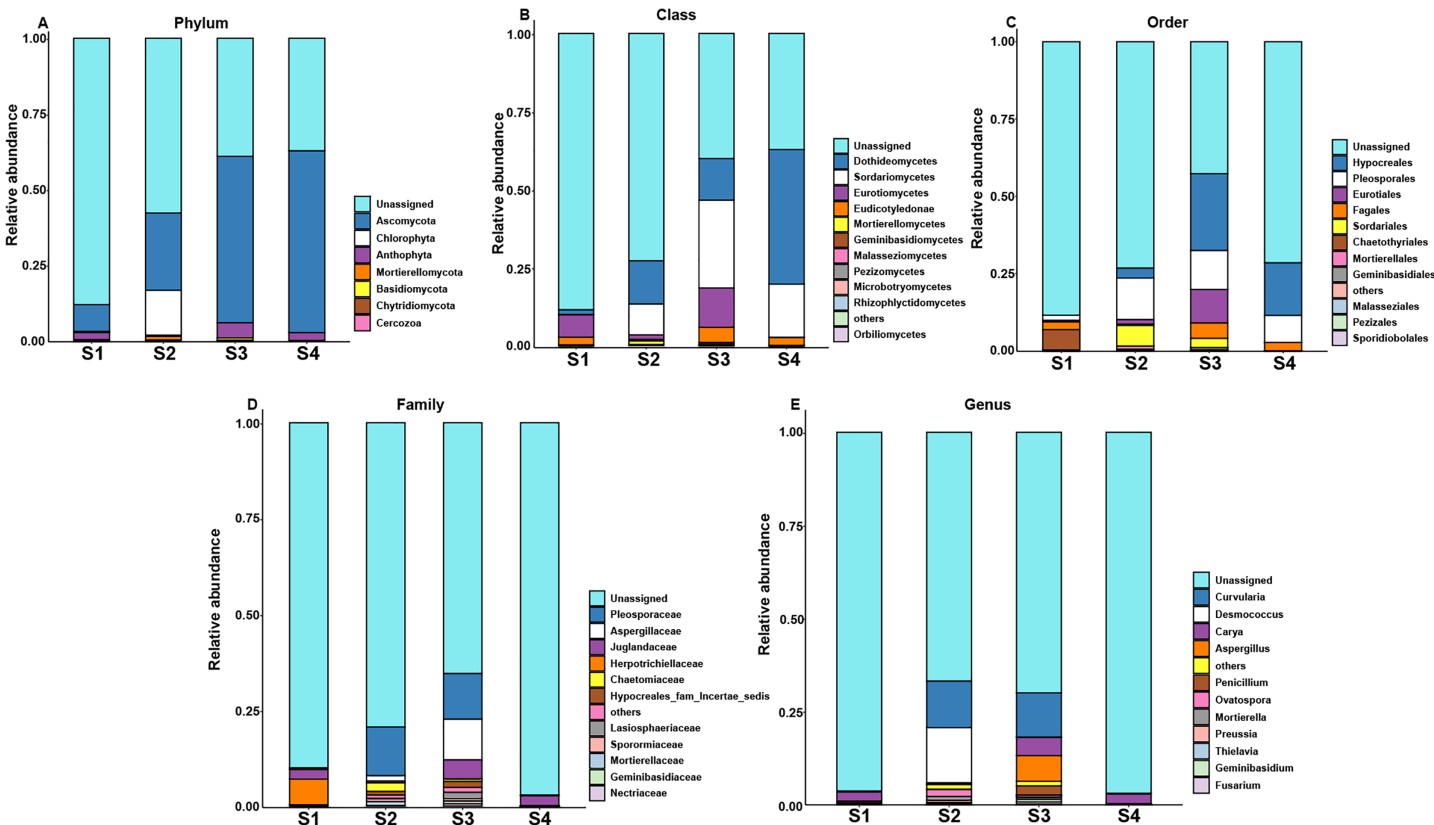

**Figure 3** Top 10 relative abundances of fungal communities classified at different germination stages of pecan seeds as: (A) phylum, (B) class, (C) order, (D) family and (E) genus.

(35.6% of the total ASVs) were shared between them (Fig. 4B) and the S3 stage shared highest number of fungal ASVs (37 OTUs, which is 20.6% of the total ASVs). To further analyze the core microbiome of the pecan seeds on the phylum level, the Venn network diagram was used because it can be more clearly to evaluate the distribution of ASVs among different stages. The common core bacterial microbiome of pecan seed consisted of members of *Cyanobacteria*, *Proteobacteria*, and *Actinobacteria* (Fig. 4C). But in fungal communities, the common core microbiome is *Anthophyta* and *Ascomycota* (Fig. 4D).

To further explore the effects of different germination stages of pecan seed on microbial communities, we analyzed bacterial and fungal communities using samples subjected to S1, S3 and S4 treatments, which affected bacterial and fungal diversity most strongly. The ternary plot showed that high-abundance ASVs belonging to *Proteobacteria* and *Actinobacteria* phyla were detected upon S3 and S4 treatments, but the relative abundances of these ASVs in the S1 were extremely low. Moreover, comparing the relative abundances of most ASVs in S3 and S4 stages, S3 is lower than S4 (Fig. 5A). Then, samples treated with S1, S3 and S4 were used to further study the fungal community changes. In the fungal community, ASVs from *Ascomycota* phylum were observed to have a high abundance in the S3 and S4 treatments, but the relative abundance of most ASVs in the S1 group was lower (Fig. 5B).

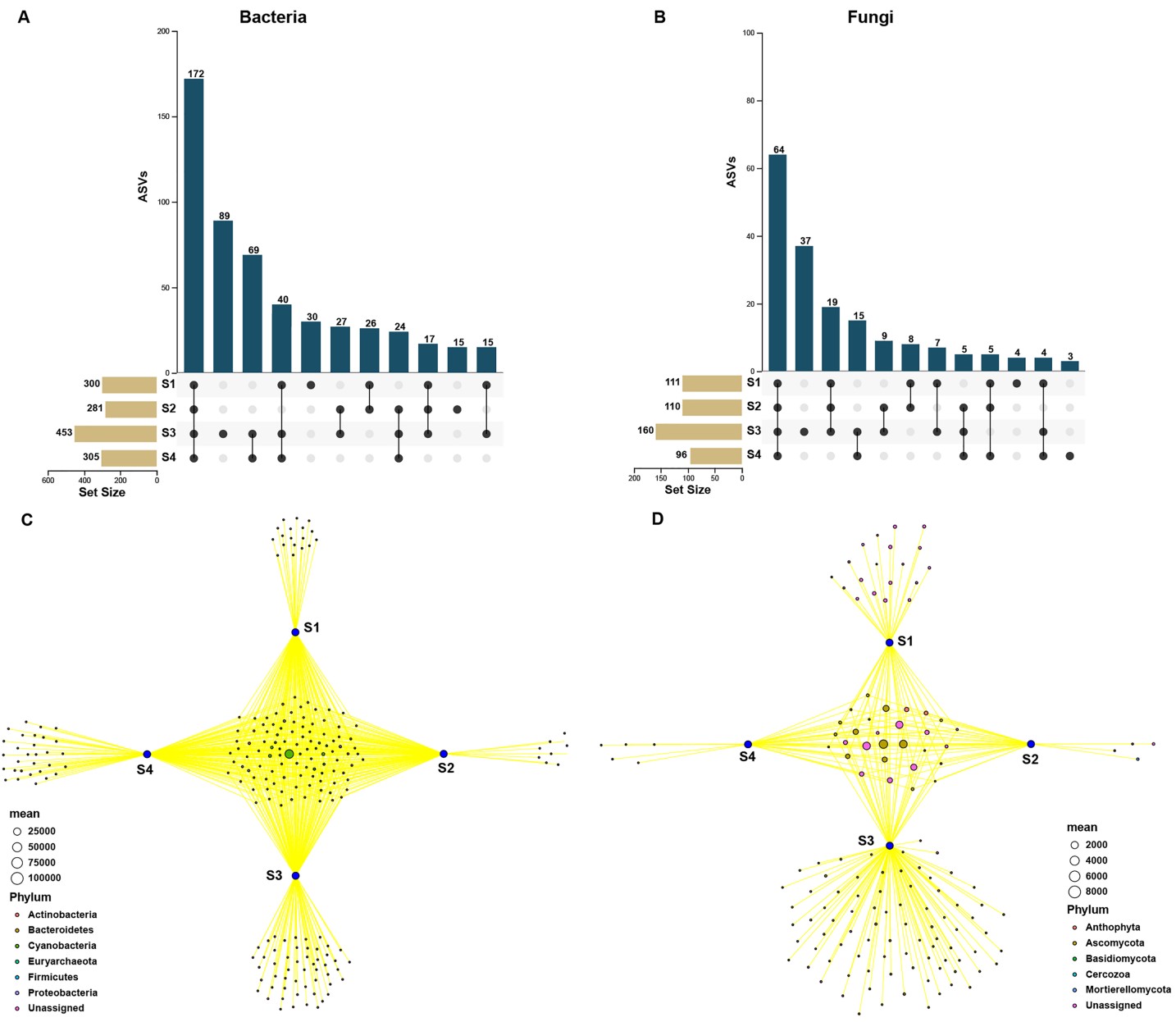

**Figure 4 Upset and Venn network diagram.** Upset diagram representing (A) bacterial and (B) fungal by ASVs associated with the pecan seed microbiome; (C and D) Venn network diagram, with nodes (ASVs) colored according to each of the main ecological clusters: (C) bacterial, (D) fungal.

## Biomarkers changes in the relative abundances of different germination stages in pecan seeds

In study, linear discriminant analysis (LDA) and effect size (LEfSe) analysis were conducted to identify bacterial and fungal communities as biomarkers (Fig. 6). Compared to the seed bacterial community (two classes, five orders, six families, nine genera and nine species), and we found significantly richer biomarker species in the fungal communities (five classes, four orders, nine families, 10 genera and 12 species) of pecan seeds in different

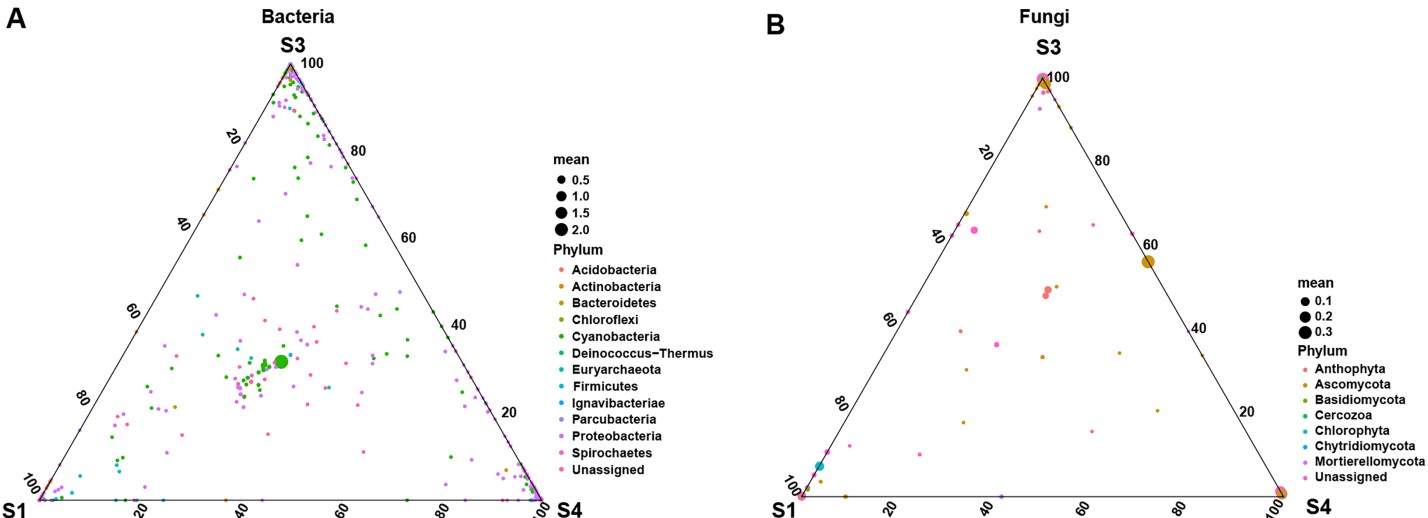

**Figure 5** Ternary plot of (A) bacterial ASV distribution for S1, S3, and S4 stages and (B) fungal ASV distribution for S1, S3, and S4 stages. Each point corresponds to an ASV. The position represents the relative abundance with respect to each treatment, the size represents the average across all three treatments, and the color represents bacterial or fungal phylum.

germination stages. All taxonomic levels of seed bacteria were associated with 10 taxa derived from S2, 29 taxa derived from S3, and 14 taxa derived from S4. Among them, *Proteobacteria*, *Gamma proteobacteria* and *Cyanobacteria* were identified as highly abundant biomarkers in the bacterial community. Among all taxonomic levels of seed fungi, 28 biomarkers were related to S1 and 18 were related to S2, 43 biomarkers were related to S3, 12 biomarkers related to S4 germination stage. *Ascomycota* and *Sordariomycetes* were significantly abundant in the seed fungal community.

## DISCUSSION

The germination of pecan seeds affected the diversity and abundance of internal microorganisms. In this study, through the analysis of the microbial community diversity and community composition in four different germination stages of pecan seeds, we found that the microbial richness and diversity were the highest in the S3 (seed endocarp break) stage. The diversity and abundance of bacteria and the abundance of fungi gradually increased from the water absorption stage to the endocarp rupture, and decreased when the radicle protruded, and speculated that this trend shift was related to the endocarp dehiscence. We speculated that this change of trend was related to the dehiscence of the endocarp. Previous studies have shown that the increase in the relative abundance of symbiotic microorganisms during seed germination and emergence is due to the spermosphere formed during seed germination providing efficient nutrients for the growth of microbial groups in seeds, which originally existed in dormant forms inside the seeds. Microbes, which break dormancy with nutrients released by germinating seeds (*Barret et al., 2015*; *Kwan et al., 2015*; *Roberts et al., 2000*; *Torres-Cortés et al., 2018*).

Endophytic bacterial biomarkers such as *Proteobacteria*, *Gamma proteobacteria* and *Cyanobacteria* were identified at different germination stages of pecan seeds by LEfSe analysis, and we speculate that they play an important role in the seed germination process.

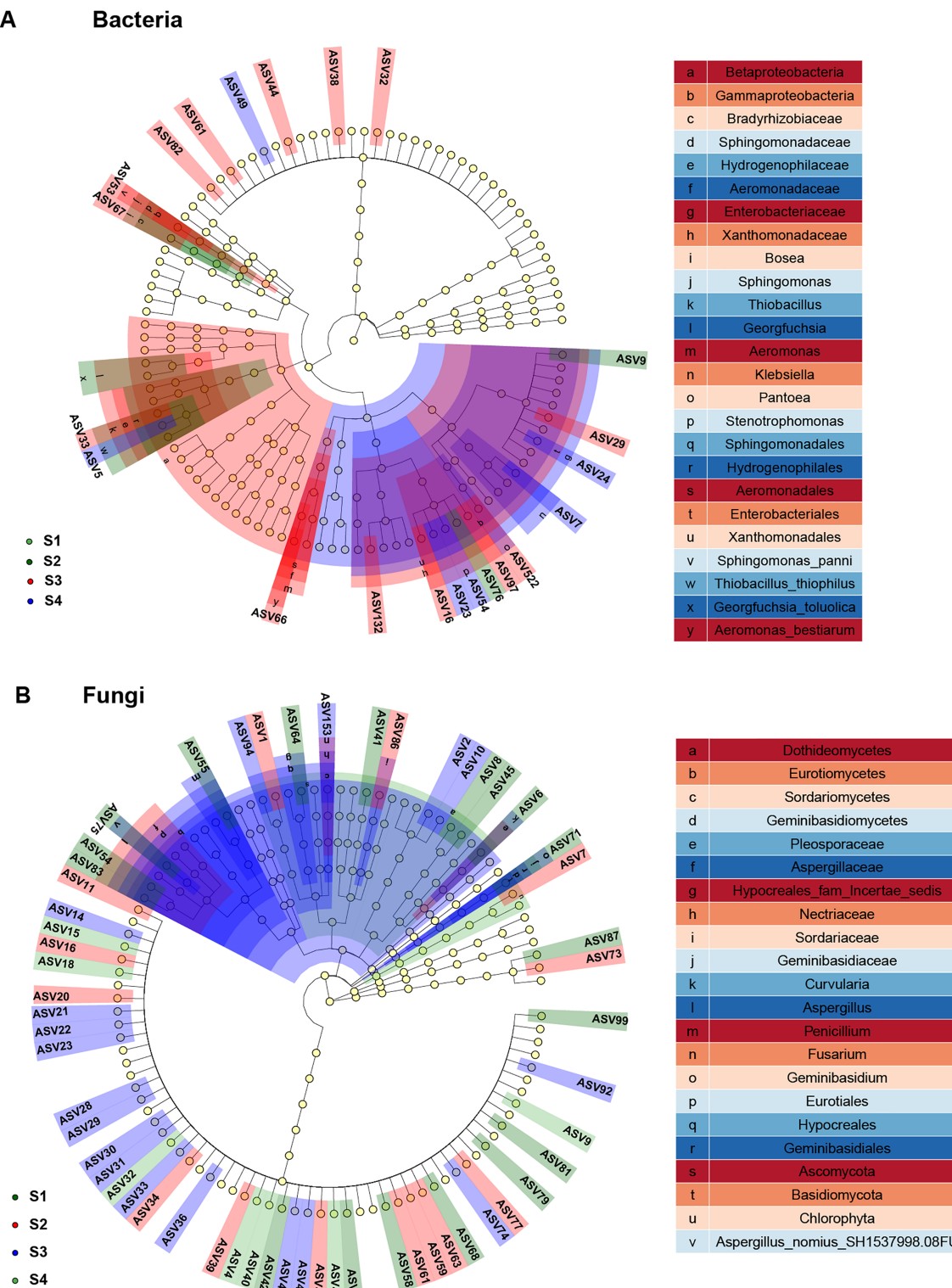

**Figure 6 Linear discriminant analysis eûect size (LEfSe) of the bacterial and fungal communities with an LDA score higher than 4.0 and P-values < 0.05.** (A) Bacterial communities. (B) fungal communities.

Among them, *Cyanobacteria* and *Proteobacteria* have high abundance, which were similar to many research results (*Chen et al., 2018*; *Liu et al., 2013*; *Zhang et al., 2012*). During seed germination, *Cyanobacteria* and *Proteobacteria* constituted almost the entire bacterial community and were also identified as core and specific microbiomes. Although the relative abundance of *Cyanobacteria* decreased during S3 and S4, it always maintained the highest relative abundance. The relative abundance of *Proteobacteria* increased rapidly in the S3 stage and remained in the S4 stage. *Cyanobacteria* has been proven to be a beneficial microorganism that can improve plant nutrient uptake capacity, promote growth, and enhance plant tolerance to stress (*Rady, Taha & Kusvuran, 2018*). In addition, *Cyanobacteria* also had a certain promotion effect on seed germination and growth. For example, *Cyanobacteria*-treated maize seeds and found that it has a good effect on promoting maize growth, photosynthesis and anti-Cd toxicity (*Seifikalhor, Hassani & Aliniaeifard, 2020*). *Cyanobacteria* treatment increased sunflower seed yield and growth (*Abdel-Hafeez, El-Mageed & Rady, 2019*). However, the effects of bacteria *Proteobacteria* and *Gamma proteobacteria* from *Proteobacteria* on seed germination have been rarely reported, and further research is needed.

Endophytic fungal biomarkers (*Ascomycota* and *Sordariomycetes*) were identified at different germination stages of Pecan seeds. *Sordariomycetes* is a class belonging to the phylum *Ascomycota*, and its relative abundance increases gradually after the seeds absorb water until the endocarp of pecan seeds ruptures. Therefore, we speculate that *Sordariomycetes* and *Ascomycota* are played an important role in pecan seed germination. *Sordariomycetes* is the main dominant class of endophytic fungal communities in plants with antipathogenic activity (*Ettinger & Eisen, 2020*; *Wang et al., 2019*; *Zhang et al., 2021*). *Ascomycota* is the most abundant phylum annotated among endophytic fungi at each stage of pecan seed germination, and it is also one of the core and specific microbiomes of endophytic fungi identified in this study. However, no studies have thoroughly elucidated the effects of the seed endophytic fungi Ascomycota and *Sordariomycetes* on germination.

Seed germination is a complex process in which initially dormant seeds undergo a series of active changes in physiological state (*Huang et al., 2018*). The abundant microorganisms inside the seeds improve the access to nutrients and enhance the ability of seeds to resist pathogens and abiotic stresses by interacting with seeds (*Dai et al., 2020*; *Morella, Zhang & Koskella, 2019*; *Seifikalhor, Hassani & Aliniaeifard, 2020*). These positive effects on various physiological activities of the host allow the host to adapt to changing environmental conditions, especially in early life stages (*War et al., 2023*). Seed endophytic microbes can promote seed germination and seedling growth, and the transition from seed to shoot is marked by changes in bacterial and fungal community composition and increased dominance (*Verma, Kharwar & White, 2019*; *Walitang et al., 2017*). Therefore, it is of great significance to explore the relationship between seed germination stages and endophytic microorganisms. In this study, high-throughput sequencing of pecan seeds at different germination stages was carried out, and the diversity and abundance changes of endophytic microorganisms in different germination stages of pecan seeds were studied by combining various analysis methods, and the core microorganisms and microbial markers in the seed germination process were found. It will help to explore the interaction

mechanism between seeds and microorganisms at different germination stages in the future, and lay a foundation for the development and research of endophytes in pecan seeds in the future.

## CONCLUSIONS

The diversity and composition of microorganisms were varied in different germination stages, and the microbial richness and diversity were the highest in the S3 (seed endocarp break) stage. It was speculated that the change of endophytes in pecan seeds was related to the germination stage. Also, we found the key microorganisms and microbial markers in the seed germination process, in addition to studying changes in endophytic microorganisms during different stages of seed germination. Thus, this study will help to explore the interaction mechanism between pecan seeds and endophytes in different germination processes, and provide materials for the research and development of pecan seed endophytes.

### Funding
This research work was funded by the National Natural Science Foundation of China (Grant No. 32001310), the Anhui Provincial Educational Foundation (2022AH030111 and 2023AH051628) and the Chuzhou University Start-up Foundation for Research (2020qd33, 2021qd05). The funders had no role in study design, data collection and analysis, decision to publish, or preparation of the manuscript.

### Grant Disclosures
The following grant information was disclosed by the authors:
National Natural Science Foundation: 32001310.
Anhui Provincial Educational Foundation: 2022AH030111 and 2023AH051628.
Chuzhou University Start-up Foundation: 2020qd33 and 2021qd05.

### Competing Interests
The authors declare that they have no competing interests.

### Author Contributions
- Jia Liu performed the experiments, analyzed the data, prepared figures and/or tables, authored or reviewed drafts of the article, and approved the final draft.
- Sumei Qiu analyzed the data, prepared figures and/or tables, and approved the final draft.
- Liping Yang analyzed the data, prepared figures and/or tables, and approved the final draft.
- Can Yang analyzed the data, prepared figures and/or tables, and approved the final draft.
- Tingting Xue conceived and designed the experiments, performed the experiments, authored or reviewed drafts of the article, and approved the final draft.

- Yingdan Yuan conceived and designed the experiments, performed the experiments, authored or reviewed drafts of the article, and approved the final draft.

## Data Availability

The raw Illumina sequence data was stored for bacterial 16S and ITS fungal data at NCBI Sequence Read: PRJNA940091 (Bacterial) and PRJNA940387 (Fungal).

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
