# Peer review of "Germination of pecan seeds changes the microbial community"

_PeerJ, doi:10.7717/peerj.16619_

## Round 0.1 · original submission · Major Revisions

The reviewers have provided some comments that the authors need to address. Please respond to the reviewers' comments with careful consideration.

**Language Note:** The review process has identified that the English language must be improved. PeerJ can provide language editing services - please contact us at [email protected] for pricing (be sure to provide your manuscript number and title). Alternatively, you should make your own arrangements to improve the language quality and provide details in your response letter. – PeerJ Staff

·

Basic reporting

• There are some grammatical mistakes in the introduction, background information and review literature (line 17, 23, 24, 25, 26, 32, 34, 55, 56, 58, 60, 73, 74).
• Common name with scientific name of the plant species should be given at least one place in whole manuscript (line 62).
• Plant name should be in italic font (line 60).
• Incomplete sentences should not be there in any section of the manuscript.
• References cited in the manuscript should follow the journal pattern while giving in running text or in reference section; Italic font use for citing reference in running text (yellow coloured); Scientific name of the crops/plants should be italic (in reference section); Mention page numbers (line 309, 322, 325, 336, 341, 348, 350, 382, 411, 423, 425, 428).
• Title of the all figure should be at the base of figure; DPI of the images should be 600 or more for clear presentation
• Give statistical test at the base of table, used for data comparison

Experimental design

• Authors of the manuscript conducted an original research on “Changes of microbial community during germination of pecan seeds”; it has wide scope in the field of seed science for improving seed germination, plant growth and development. Research problem is well defined by the authors.
• Is there any protocol developed by scientists regarding seed sample collection method for such kind of studies? If yes then give reference.
• DNA extraction, amplification and sequencing method is appropriate for the present study. Sequence analysis for bacteria and fungi, annotation done with appropriate standards. Analysis followed appropriate statistical indices/methods with the help of suitable applications.

Validity of the findings

• Give full form of short forms at least one used in the manuscript; give full form of ASVs (line 132)
• Need corrections noted in soft copy of the manuscript (line 135 to 138; 141, 155, 157, 175, 181, 187, 190, 193, 195, 196, 200, 203, 207, 219, 222, 230, 233, 235, 236, 238, 263)
• Explain, why the number of genera were higher than number of species? (line 151 and 163)
• Sentence should be grammatically correct and complete (line 228 & 229; 231 to 233)

Additional comments

• There are many grammatical mistakes in the manuscript, needs to rectify (see the comments given on the manuscript pdf file)
• Results explained and discussed in the manuscript are well inferenced and concluded the outcomes of the research
• Rather than that Research and Discussion section needs improvement

·

Basic reporting

• Some grammatical mistakes in the abstract, introduction, material and methods, results, discussion and conclusion sections of the manuscript, which is highlighted in the whole manuscript and also seen the comments box for suggestions.
• See the key words and plz corrected.
• Write scientific name of microbiota and plants should be in italic font in whole manuscript.
• In Introduction section please write the hypothesis of the research work clearly.
• In Material and Methods section, the title of the sub heading changes as per the running matter of your manuscript – Please incorporate as per suggestion given on manuscript.
• Write the abbreviations at least one time of all short form used first time in the manuscript in bracket, like PCA (Principal Component Analysis) etc.
• In reference section – Follow the journal formate carefully, scientific name of the micobiota and plants should be in italic font (in whole reference section) and check the comments in reference section of the manuscript.
• Rewrite the titles of figure as per suggestion given on manuscript clearly. Also requested to enhance the image pixel or quality for clear vision.
• Incorporate the suggestions given on table 1 also.

Experimental design

All suggestions mentioned in manuscript

Validity of the findings

Please see the suggestions highlighted in the whole manuscript and incorporated accordingly

Additional comments

I found some grammatical mistakes in manuscript,
Some sentences are rewritten please.

---

## Round 0.2 · Minor Revisions

The manuscript has been significantly improved; however, there are a few minor comments that the authors should address to further enhance the manuscript.

Please respond to those comments carefully

·

Basic reporting

• There are few corrections in the introduction, background information and review literature (line 17, 19, 24, 48, 66, 190, 238).
• Phyla name should be in italic font (line 159, 188, 192).
• Family name should not in italic font (line 60, 167)
• DPI of the images should be 600 or more for clear presentation
• What does the ± values indicates in table? It should mention at the end of table. (Is it SD or SE?)

Experimental design

Nicely explained, haven't corrections

Validity of the findings

• Explain, why the number of genera were higher than number of species?

Additional comments

• Needs to rectify the corrections (see the comments given on the manuscript pdf file)
• Results explained and discussed in the manuscript are well inferenced and concluded the outcomes of the research

·

Basic reporting

Author corrected all necessary corrections made on manuscript, but some minor corrections also suggested for manuscript improvement.
Author also requested to see suggestions made on manuscript and corrected carefully, because I observed that in some sentences we and our type words are seen again.

Experimental design

Authors have incorporated properly

Validity of the findings

Author written nicely and corrected

Additional comments

See comments mentioned in pdf file it self
Table 1 corrected as suggested
In Table 1 and Colum 2 write sample ID in symbol form like S1, S2, S3 and S4 and just below the table write NOTE : S1 - ............................., S2 - .........................., S3 - ........................... and S4 - ..................... (Write full form of S1, S2, S3 and S4)

---

## Round 0.3 · accepted · Accept

The current manuscript meets the journal standards; thus it can be accepted for publication.